# Spatiotemporal persistence of multiple, diverse clades and toxins of *Corynebacterium diphtheriae*

Robert C. Will [1,10], Thandavarayan Ramamurthy[2,10], Naresh Chand Sharma[3,10], Balaji Veeraraghavan[4], Lucky Sangal[5], Pradeep Haldar[6], Agila Kumari Pragasam[4], Karthick Vasudevan[4], Dhirendra Kumar [2,3], Bhabatosh Das [2], Eva Heinz [7], Vyacheslav Melnikov [8], Stephen Baker [1], Vartul Sangal [9], Gordon Dougan[1] & Ankur Mutreja [1,2✉]

Diphtheria is a respiratory disease caused by the bacterium *Corynebacterium diphtheriae*. Although the development of a toxin-based vaccine in the 1930s has allowed a high level of control over the disease, cases have increased in recent years. Here, we describe the genomic variation of 502 *C. diphtheriae* isolates across 16 countries and territories over 122 years. We generate a core gene phylogeny and determine the presence of antimicrobial resistance genes and variation within the *tox* gene of 291 *tox*+ isolates. Numerous, highly diverse clusters of *C. diphtheriae* are observed across the phylogeny, each containing isolates from multiple countries, regions and time of isolation. The number of antimicrobial resistance genes, as well as the breadth of antibiotic resistance, is substantially greater in the last decade than ever before. We identified and analysed 18 *tox* gene variants, with mutations estimated to be of medium to high structural impact.

[1] Department of Medicine, Cambridge Institute of Therapeutic Immunology & Infectious Disease (CITIID), University of Cambridge, Cambridge, UK. [2] Translational Health Science and Technology Institute, Faridabad, India. [3] Maharishi Valmiki Infectious Diseases Hospital, Delhi, India. [4] Department of Clinical Microbiology, Christian Medical College, Vellore, Tamil Nadu, India. [5] World Health Organization, New Delhi, India. [6] Ministry of Health and Family Welfare, Govt. of India, New Delhi, India. [7] Department of Vector Biology, Liverpool School of Tropical Medicine, Liverpool, UK. [8] Gabrichevsky Research Institute for Epidemiology and Microbiology, Moscow, Russia. [9] Faculty of Health and Life Sciences, Northumbria University, Newcastle upon Tyne, UK. [10] These authors contributed equally: Robert C. Will, Thandavarayan Ramamurthy, Naresh Chand Sharma. ✉email: am872@medschl.cam.ac.uk

Diphtheria, traditionally regarded as a toxin-driven disease of the respiratory system, usually begins with angina or tonsillitis symptoms, sore throat and mild fever. However, the clinical picture can quickly escalate leading to death if the disease is not treated[1]. A white-grey pseudomembrane over the pharynx, larynx and tonsils is considered stereotypical to the disease, as is a swollen bull neck, although these symptoms are not displayed in all the infected cases[2,3]. Once a common cause of infection, diphtheria has been vaccine preventable for decades and is now rarely observed in high income countries[2,4]. In low and middle income countries, however, diphtheria is still of concern as it can cause sporadic infections or outbreaks in unvaccinated and partially vaccinated communities[3,5]. Any drop in the levels of diphtheria vaccine coverage could potentially lead to an opportunistic return of this communicable disease, as already being reported for other vaccine-preventable diseases like measles and pertussis[6–9]. The number of diphtheria cases reported globally has followed a gradually increasing trend in recent years, with the cases in 2018 (16,651) being over double the 1996–2017 average (8105)[10].

Diphtheria is primarily caused by toxigenic *Corynebacterium diphtheriae*, that colonise the upper respiratory tract[11]. The route of transmission between humans is likely through droplets contaminated with *C. diphtheriae*[12]. The diphtheria toxin is encoded on a *tox*$^+$ corynephage in *C. diphtheriae*. Related species including *C. ulcerans* and *C. pseudotuberculosis* can also be lysogenised by this phage type, and such strains are also associated with a clinical disease almost identical to diphtheria[13–16]. In addition, non-toxigenic *C. diphtheriae* can cause disease, often in the form of systemic infections[17,18]. Non-toxigenic but toxin gene-bearing (NTTB) *C. diphtheriae* have also been recorded[19,20]. Since the universal diphtheria vaccine is a partially pure toxoid formulation, it may not be as effective for preventing NTTB led infections. Acute diphtheria is usually treated with anti-diphtheria toxin serum, alongside a course of antibiotics[21]. While *C. diphtheriae* resistant to antibiotics have been reported, the extent of such resistance in this pathogen remains largely unknown[2,22–24]. A complete reference genome for *C. diphtheriae* has been available for almost two decades[25]. Subsequent genomic analysis has predominantly focused on small, geographically clustered isolates, alongside reference genomes to provide a context[26–29].

In this work we seek to understand the genomic dynamics of *C. diphtheriae* more widely with a detailed focus on India, where over 50% of globally reported cases occurred in 2018[10]. We interrogate the genomes of a large collection of *C. diphtheriae* isolates to determine their phylogenetic structure, assess the presence of antimicrobial resistance (AMR) genes and assess toxin variation, which is the key target of the current diphtheria vaccine.

## Results

**Genomic insight into 122 years (1896–2018) of *C. diphtheria*.** A collection of 502 *C. diphtheriae* genomes was established by sequencing 61 novel Indian isolates and combining these with 441 publicly available genomes. The isolates in our study cover 16 countries and territories and they were collected across a period of 122 years. As India has reported the highest number of cases in the last few years, a separate Indian subset totalling 122 genomes with two isolates from 1973 and the rest collected between 2015 and 2018 (including the 61 novel isolates), was created and analysed.

An initial analysis of the total 502 *C. diphtheriae* genomes showed that, as previously reported, it is a genetically diverse species. An initial mapping analysis revealed evidence for extensive recombination across the whole species, not conducive to an accurate phylogeny (Supplementary Fig. 1). 100 Multilocus sequence types (MLST) were determined, with 73 isolates designated as 'novel STs'. The core gene list was calculated to be 1035 genes with an extended pan gene list accumulating to 23,447 genes. The Indian subset had 1367 core genes but a lower number of 7436 genes in the pan gene list across all 122 genomes (Supplementary Fig. 2).

Figure 1A displays the outcome of the phylogenetic analysis of all 502 genomes included in our study. Built from a 49,454 nucleotide long core gene single nucleotide polymorphism (SNP) alignment, the phylogenetic tree in Fig. 1A indicated the presence of several clusters across the collection, each containing isolates from wide temporal and spatial ranges. Within these clusters, single monophyletic groups were readily identifiable, and these predominantly showed strong geographic and temporal association. Isolates within a particular monophyletic group shared a single ST, although some individual isolates within these groups were also 'novel STs'. One subclade spread extensively to cover multiple neighbouring European countries—predominantly Belarus and Germany—and persisted over a large period of time (marked by a blue star). Isolates within this lineage were isolated as early as 1996 and persisted until at least 2017 when isolates were found in Germany. Clusters contained groups isolated from multiple continents, most commonly Asia and Europe. This clearly indicates that *C. diphtheriae* has been established in the human population for at least over a century and has spread, potentially by population movements, across time and space. This also shows that there are multiple distinct clonal populations circulating in the same geographical setting[1,29].

Within India, isolates from individual monophyletic groups were from multiple states, spanning both Northern and Southern India (Fig. 1B). The closely related monophyletic groups of similar origins suggest clonal outbreaks occurring opportunistically, perhaps emerging from commensalism when favourable factors occurred.

**tox gene diversity.** Diphtheria toxin is the main virulence-associated factor in the disease. In total, 18 allelic variants of the *tox* gene were found across the 291 *tox*$^+$ isolates, which make 58% of the total collection of 502. Of these 18 allelic types, eight were found to contain non-synonymous SNP changes with types 5, 8, 14, 15, 17 and 18 each containing one amino acid substitution. Toxin type 7 contained two amino acid substitutions, one of which was shared with type 8. Group 13 instead harboured a deletion, previously reported to result in the strain being NTTB. The impact of these non-synonymous mutations on the protein structure was estimated using PHYRE2 and SuSPect, and were plotted onto the diphtheria toxin protein model 1XDT (https://www.rcsb.org/structure/1xdt) from the Protein Data bank using UCSF ChimeraX (Fig. 2)[30–34]. Groups 13 deletion and group 14 mutations were both present in the signal sequence of the gene, and thus could not be mapped to the protein structure. Groups 7 and 8 shared mutation was estimated to have low impact, the second mutation in group 7 as well as groups 5, 15, 18 were calculated as moderate impact mutation, while group 17 mutation was calculated as a high impact mutation. Notably, the site of groups 15, 17 and 18 mutations had a much higher average mutation impact, with many of the potential amino acid substitutions showing a high risk of mutations impacting on the toxin structure. The proportions of the *tox* allelic variants found in each decade is shown in Fig. 1C. The most common toxin variant was type 16 (49.5% of toxigenic isolates), with 4, 6, 8, 11 and 15 being the next most common alleles. Type 16 also includes the *tox* variant found in the vaccine strain, PW8[35]. Three *C.*

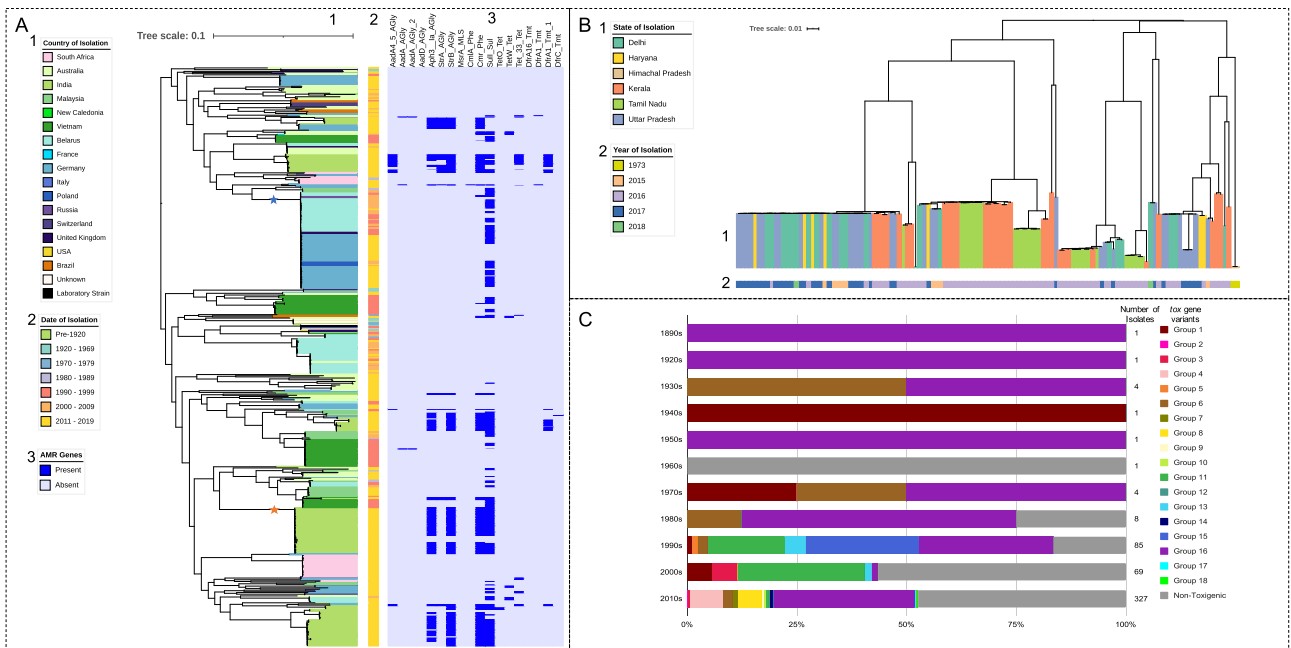

**Fig. 1 Global and Indian core gene phylogenies of *Corynebacterium diphtheriae* and *tox* gene variants by decade. A** Maximum likelihood phylogenetic tree based on the core gene single nucleotide polymorphisms from the 502 global Corynebacterium diphtheriae genome collection. The country of isolation (1), decade of isolation (2), and AMR gene presence/absence heatmap made using ARIBA[50] (3), are all annotated. Most monophyletic groups within the tree represent only one country and decade, with the major exception of the large Belarussian/German dominated group, marked with a blue star. Both the blue and orange stars highlight groups used for BEAST analysis. **B** The core gene maximum likelihood phylogeny of only the 122 Indian isolates, coloured by state (1) and year of isolation (2). **C** The proportion of the 18 tox gene variants found across 291 tox+ and 211 non-toxigenic isolates per decade, with the number of isolates per decade shown.

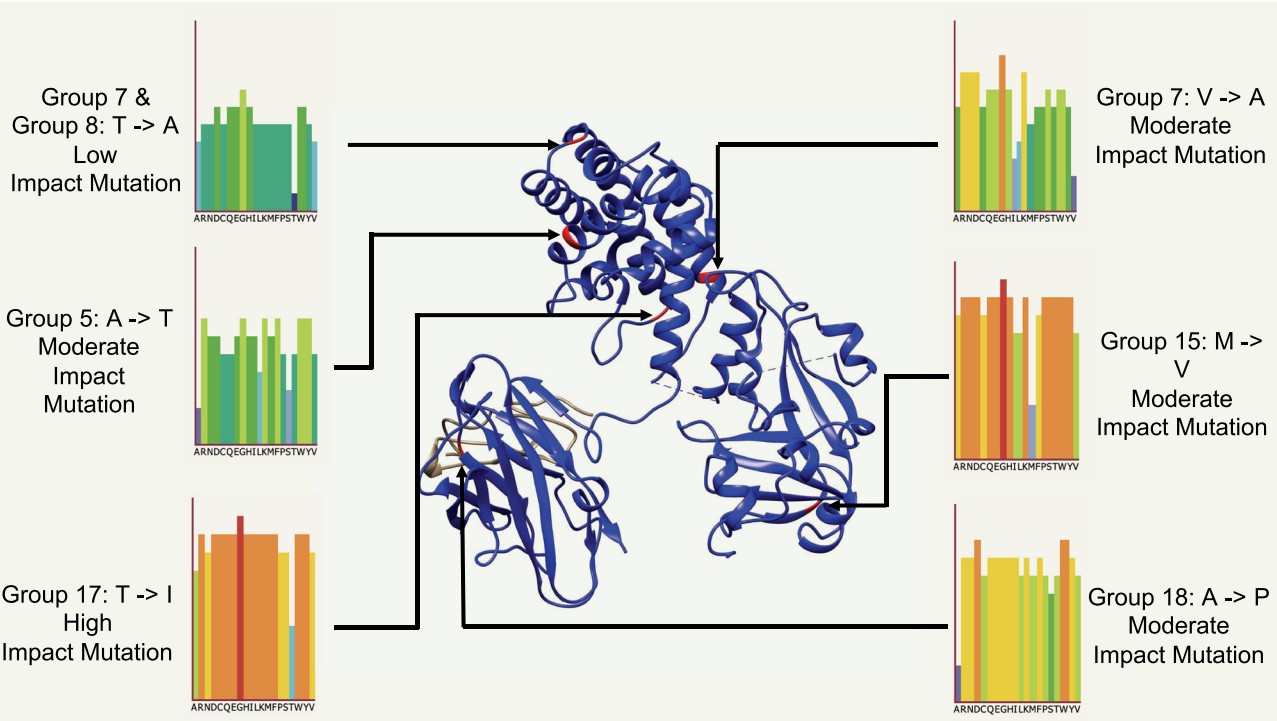

**Fig. 2 Six non-synonymous mutations plotted onto diphtheria toxin model 1XDT (https://www.rcsb.org/structure/1xdt) from the Protein Data Bank using PHYRE2.** The impact of these mutations is estimated by SuSPect, with a gradient per mutation of low (dark blue) to high (orange/red).

*diphtheriae* isolates carried two identical copies of the *tox* gene, two type 7 isolated in Germany and Switzerland, and one type 16 isolated in Germany. *tox* variant diversity significantly increases by decade ($r$ (9) = 0.70, $p = 0.02$), although this may be because

of unavoidable bias of sampling. Supplementary Figs. 3, 4 show the breakdown of *tox* types by country of origin and year of isolation. Non-toxigenic isolates were much more common in mainland Europe, Brazil and Australia than in India and

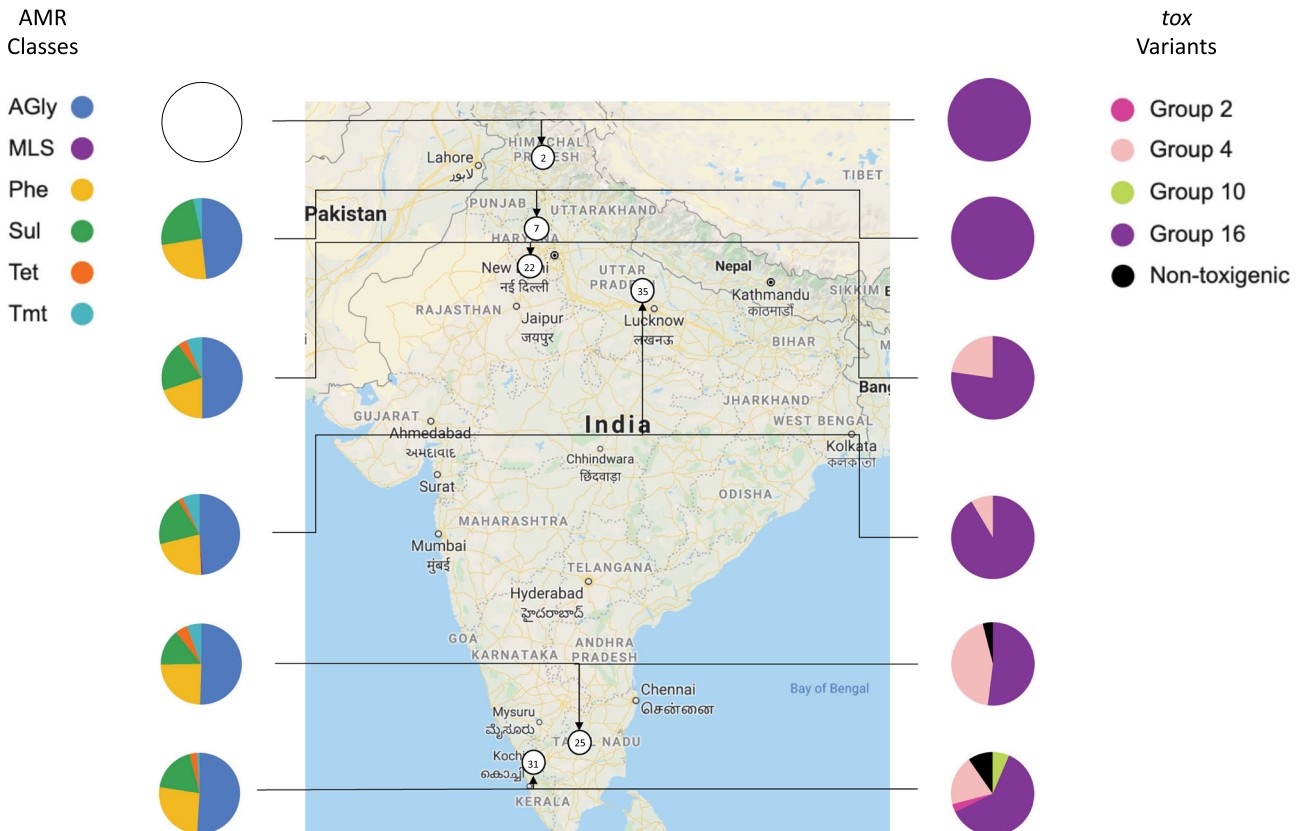

**Fig. 3 The antimicrobial resistance (AMR) gene and *tox* gene variant proportions represented in the 122 *Corynebacterium diphtheriae* genomes from India at state level.** The number of isolates from the six states sampled are shown in circles on the map. The AMR genes are coloured by the classes of antibiotic the genes offer resistance to. AGly (blue) aminoglycosides, MLS (purple) macrolide–lincosamide–streptogramin, Phe (yellow) phenicols, Sul (green) sulfonamides, Tet (orange) tetracyclines, Tmt (light blue) trimethoprim. The map was taken from Google Maps[47].

Southeast Asia, as well as being more common in more recently isolated genomes (non-toxigenic isolates per decade ($r$ (9) = 0.60, $p$ = 0.05). The difference between the number of non-toxigenic and toxigenic isolates in HICs and LMICs was significantly different ($r$ (1) = 202.67, $p$ < 0.001). Group 16 was the predominant variant across South Asia, while being much less common outside the region. Group 16 was also the predominant variant in recent years, despite the rise in non-toxigenic isolates. Some *tox* variants were present only within a single monophyletic group and location e.g. type 8, only present in 18 South African isolates, and groups 17 and 18, only present in one isolate each from Australia. Others including types 4 and 16 were present throughout the phylogeny, across numerous clusters and groups.

Figure 3 shows the proportion of each *tox* type per state across India. Four of the eighteen toxin types were represented among the Indian isolates, with Group 16 still being the dominant variant. Types 2 and 11 were only present in Kerala. The only non-toxigenic Indian isolates were found in the South, in Tamil Nadu and Kerala.

**Antimicrobial resistance in *C. diphtheria*.** The AMR gene portfolio (Fig. 1A) and the average number of AMR genes per decade (Supplementary Fig. 5) were evaluated. More recently isolated genomes (including those from India) showed a disproportionately high number of AMR genes, with a significant positive correlation between the average number of AMR genes per genome and the decade of isolation ($r$ (9) = 0.68, $p$ = 0.02). Similar resistance profiles were found across a small number of recent Swiss and German isolates. The sulphonamide resistance

gene (*sulI*) was the most common, being consistently found in Indian and sporadically in other Asian and European isolates. Aminoglycoside (*aadA4, aph3_Ia, strA* and *strB*), chloramphenicol (*cmr*) and trimethoprim (*dfrA1*) resistance genes were also present in most recent Indian isolates. Only one isolate (from India) was found to contain a macrolide resistance-encoding gene (*msrA*), while no isolates harboured β-lactam resistance genes. Isolates from the decade of 2010 to 2019 shows the highest average number of AMR genes per genome, almost four times as many genes on average than in the next highest decade; the 1990s (Supplementary Fig. 5). The 2010s also showed more variation in the classes resisted (six), compared to other decades.

Within the Indian isolates, *C. diphtheriae* from Himachal Pradesh (isolated in 1973) had no detectable AMR genes (Fig. 3). Isolates from Haryana harboured genes encoding resistance to four classes of antibiotic, whereas isolates from Delhi, Kerala and Tamil Nadu had genes for resistance to five. Isolates from the most populous state of India, Uttar Pradesh, showed resistance to six classes, including the only isolate of our study with macrolide resistance. Five isolates from Kerala (all isolated in 2016) and one from Uttar Pradesh (isolated in 2017) showed no AMR genes present in their genomes. Phenotypic AMR testing was carried out for the 61 novel Indian isolates and these results align well with the genes found in silico (Supplementary Data 1).

**Time scaled phylogenetic analysis.** The European and Indian groups (marked with blue and orange stars respectively on Fig. 1A) are shown in Supplementary Figs. 6, 7. BEAST estimates

the European group's most recent shared ancestor was in September 1983. Three clades are present within the main European clade with the first spanning March 1987–2008. The second and third clades are estimated to have diverged in July 1985 with the former present from September 1988 to 2010 and the latter estimated to being in circulation between April 1990 and 2017. The ratio of pre- and post-recombination removal SNPs included in the alignment was 1.1 (1944:1783).

The Indian group is estimated to have shared an ancestor with the reference isolate NCTC11397 in October 1955, before diverging into two clades in February 2009. The first clade is estimated to have run from April 2012 to 2017, with the second from June 2013 to 2018. The ratio of pre- and post-recombination removal SNPs included in the alignment was 1.2 (26879:22717).

**Corynephage variation**. The maximum likelihood phylogeny of 11 toxigenic corynephages from complete *C. diphtheriae* genomes can be seen in Supplementary Fig. 8, based on a mapped alignment of 36,570 nucleotide bases. There were no major phylogenetic correlations with the country of isolation and the decade of isolation as closely related corynephages being isolated over large temporal and geographical distances. The *tox* gene variant carried by the corynephage also does not correspond with the phylogenetic structure, with groups 6 and 16 found in distantly related isolates across the phylogeny.

## Discussion

By analysing a large collection of *C. diphtheriae* genomes we identified numerous clades distributed across the globe. Individual clades contained monophyletic groups isolated from multiple countries and regions across our sampling time frame. In India, where the majority of recently reported diphtheria cases originate, numerous monophyletic groups were found across the phylogeny as several independent lineages. These sub-lineages of *C. diphtheriae* thus represent a persisting and apparently successful diverse population. This is in contrast to other bacterial pathogens such as *Mycobacterium abscessus*, which are represented by largely clonal lineages that have spread across the globe. The high level of diversity and recombination once again highlights the major challenges of creating high confidence phylogenies of *C. diphtheriae*. Core gene approaches remain the most robust methodologies for building phylogenies of large *C. diphtheriae* collections to cope with the large diversity and recombination across major clades. Mapping approaches and recombination removal have been demonstrated as possible for closely related groups, highlighting the need for more high-quality reference genomes representing these currently circulating lineages. We were able to use BEAST to create time-scaled phylogenies of individual closely related monophyletic groups, reaffirming the importance of these high-quality reference genomes. This demonstrates that time-scaled phylogenies could be used to accurately analyse individual outbreaks and support public health measures, provided a single group of *C. diphtheriae* is responsible, rather than numerous clusters concurrently as seems to be the case in India and other parts of the world.

Although our study is based on a comprehensive clinical strain collection, our study lacks data on carriage isolates from non-clinical cases, as *C. diphtheriae* is not routinely screened for in healthy individuals. Nevertheless, the spatio-temporal structure of the phylogeny suggests that carriage could be playing a significant role in the overall persistence and evolution of the *C. diphtheriae* species. Therefore, in future studies, it is important to monitor the human population for asymptomatic carriage in both vaccinated and unvaccinated individuals, especially in at-risk communities and known epidemiological hot spots.

While our data does not, at present, highlight any efficacy concern in the currently used *tox* variant type 16 based diphtheria toxoid vaccine, the continually increasing toxin diversity and prevalence of non-toxigenic strains do however forecast a real possibility of vaccine escape and anti-toxin treatment failure in future. While the current vaccine against diphtheria continues to remain largely effective, it is vital to conduct further in vitro and in vivo studies to investigate the benefit, if any, that the six variants with non-synonymous allelic mutations that impact protein structure may provide to *C. diphtheriae*, especially those estimated to have moderate and high impacts on structure. Further sequencing of *C. diphtheriae* isolates may reveal additional variant types and show the full extent of the spread of the variants catalogued by us. This evolutionary tracking is of critical importance as it will forewarn the public health agencies of any possible vaccine escape or anti-toxin treatment failure, allowing the planning of early alternate intervention. Our analysis of the corynephage genome, while preliminary, suggests the diversity in the *tox* gene is not merely a product of wider phage diversity. Due to being based on only the limited number of complete genomes that carried the corynephage, this conclusion requires further investigation as the number of public completed *C. diphtheriae* genomes increases. With a steady global increase in vaccination rates for diphtheria, the selection pressure on the toxin as the main antigen is bound to increase, and this may be the reason behind the increase in non-toxigenic isolates recorded, as well as the significantly lower proportion of the 18 *tox* variants found among higher income countries. Among the isolates from Europe, where variant 16 now makes up only a small fraction of *tox* variants found, and where non-toxigenic isolates have been more commonly reported, further research utilising this toxin diversity information is imperative, as is more detailed investigations into the mechanisms of non-toxigenic infection. Using advanced phylogenomics, our data guides public health preparedness and suggests that a plan B in hybrid toxin-based vaccine candidates and antitoxins must be pre-emptively prepared for action if a non-synonymous non-group 16 toxin carrying *C. diphtheriae* were to abruptly become prevalent.

AMR has rarely been considered as a major problem in the treatment of *C. diphtheriae*. Here, we demonstrate that recently in some parts of the world, genomes are acquiring resistance to numerous classes of antibiotics, likely driven through the overall exposure to antimicrobials and the common co-occurrence of different resistance elements on the same mobile elements. The resistance genotypes are not limited to a single country or region but present in both Asia and Europe. Despite erythromycin and penicillin being the traditionally recommended antibiotics of choice for treating confirmed cases of early-stage diphtheria, only one macrolide resistance gene was found, and no resistance to β-lactamases. This indicates that AMR acquisition in *C. diphtheriae* could be happening during its purported carriage state and in response to the exposure to antibiotics in their environment or during patient treatment against other infections. The independent acquisition of highly similar AMR genes in *C. diphtheriae* indicates recombination between these lineages and acquisition of mobile elements from other species. It could also be due to collateral selection, where pressure to acquire resistance to one antimicrobial agent may drive the advancement of resistance to other agents by its very development[36].

Although the results from our study are based on a comprehensive clinical strain collection of isolates and genomes, our study lacks data on carriage isolates, which could be playing a highly significant role in the overall evolution of the

*C. diphtheriae* species. Our phylogenomic findings strengthen the hypothesis that while the diphtheria vaccine may be effective in preventing the symptomatic infection, carriage in vaccinated individuals or asymptomatically infected individuals could be continuing to provide a suitable ecosystem for *C. diphtheriae* sustenance for prolonged periods. COVID-19 has negatively impacted childhood vaccination schedules worldwide, with diphtheria doses scheduled at 6, 10 and 14 weeks. This coupled with reported case numbers rising over the past decade, and the year 2018 showing the highest incidence in 22 years, it is more important than ever to understand this historically important disease, to prevent it from becoming a major global threat ever again in its original or a modified, better adapted, form.

## Methods

**Data collection**. Sixty-one novel isolates were identified from 3 Northern Indian states. Of these, 22 isolates were from patients in Delhi, 7 from Haryana, and 32 from Uttar Pradesh. Nine were isolated in 2015, 16 in 2016, 34 in 2017, and 2 in 2018. Genomes of all these isolates have been made publicly available, with accession numbers found in Supplementary Data 1. Novel Indian isolates were grown, and DNA extracted for sequencing. Whole Genome Sequencing was carried out using an Illumina HiSeq v4 platform at Wellcome Sanger Institute, producing short read whole genome sequences that were assembled and annotated using SPAdes assembly (v3.13.0) and Prokka annotation (v1.5)[37,38]. Publicly available genomic data and metadata were collated following a literature review, to frame a global representation. We also analysed our novel isolates within an Indian context, to allow comparison between the global picture and the country with the highest reported case numbers to the World Health Organisation. Microreact (v5.93.0) plots have been produced for both the global (https://microreact.org/project/CZaifLUuW) and the Indian collections (https://microreact.org/project/TuIdKrIfc) to aid in data sharing[39].

**Core gene phylogeny**. Due to the extremely high recombinogenic variation between the genomes of *C. diphtheriae* identified using Gubbins (v2.4.1) and Phandango (v0.9)[40,41], which is not conducive to an accurate phylogeny, we chose Roary (v3.13.0)[42] instead to investigate the phylogenetic relationships between isolates. Using annotated genomes produced by Prokka (v1.5)[38], Roary extracted the genes present in 99% of the isolates, concatenating them to produce a core gene alignment. The number of genes across the pan-genome categories was determined. SNPs were identified using SNP-sites (v2.5.1)[43] and used to produce a core gene maximum likelihood phylogenetic tree, using IQ-TREE (v1.6.10) and the inbuilt ModelFinder over 1000 pseudo-bootstrap replicates[44,45].

**Genomic analysis**. Results, alongside the spatial and temporal metadata, were annotated to the phylogenetic trees using the Interactive Tree of Life (iTOL) (v5.5.1)[46], to determine any epidemiological relatedness between the different phylogenetic and phylogeographic clades. To display the toxin variants and AMR genes per Indian state, a map was taken from Google Maps[47]. The MLST of all isolates was determined using MLSTcheck (v2.1.1706216)[48].

***tox* Gene analysis**. Variation in the *tox* gene was investigated by obtaining the gene sequence from 291 toxigenic isolates using in silico PCR[49]. Primers are available in Supplementary Table 1. The non-synonymous mutations found were plotted onto diphtheria toxin model 1XDT from the Protein Data Bank using UCSF ChimeraX (v1.1), and the risk of those mutations impacting on protein structure was calculated using PHYRE2 (v2.0) and the inbuilt SuSPect[30–34].

**Antimicrobial resistance testing**. ARIBA (v2.14.6) was used to interrogate the genome for the presence of antimicrobial resistance (AMR) genes, to investigate if resistance was now becoming a concern in *C. diphtheriae* globally[50]. Antimicrobial susceptibility test was performed using MICs gradient test (E-test, BioMèrieux, Marcy, l'Etoile, France). The E-test strips of ampicillin (AM), amoxicillin/clavulanic acid (XL), azithromycin (AZ), ciprofloxacin (CI), chloramphenicol (CL), clindamycin (CM), cefotaxime (CT), doxycycline (DC), erythromycin (EM), gentamycin (GM), imipenem (IP), linezolid (LZ), moxyfloxacin (MX), penicillin (PG), rifampicin (RI), tetracycline (TC, ceftriaxone (TX), vancomycin (VA), trimethoprime–sulphamethoxazole (SXT) on Mueller-Hinton agar plates with 5% horse blood. Corynebacterium spp. MIC interpretative standard was based on the Clinical and Laboratory Standards Institute[51].

**Statistics**. RStudio (v4.6.1) was used to carry out Pearson's product-moment correlation to determine the significance between decade of isolation and *tox* gene variety, the average number of AMR genes per genome, and the number

of non-toxigenic isolates per decade, while a chi-squared test of non-toxigenic and toxigenic isolates from HICs and LMICs[52,53].

**Time scaled phylogenetic analysis**. Two closely related clades were chosen for time-scaled phylogenetic analysis. Recombination was removed from the mapped alignments using Gubbins (v2.4.1) before using the BEAST (v1.10) package to carry out the Bayesian time-scaled phylogenetic analysis[40,54]. The Hasegawa, Kishino and Yano model (HKY) substitution model with different demographic models (Bayesian skyline, exponential and constant) was investigated. Markov chain Monte Carlo runs of 100 million generations were carried out with sampling of 20,000 generations[55]. The convergence of each run was manually inspected using Tracer (v1.7)[56]. A burn in of 20% was discarded from the runs and maximum clade credibility tree was finally generated using Treeannotator (v1.8.2)[57]. The annotated phylogenetic tree was visualised using FigTree (v1.4)[58].

**Corynephage analysis**. Corynephage diversity was investigated by mapping the 11 toxigenic complete genomes publicly available from NCBI Genbank to the corynephage sequence annotated in *C. diphtheriae* isolate NCTC 13129 using BWA (v0.7.17-r1188)[59,60]. This mapped alignment was then used to produce a maximum likelihood phylogenetic tree with IQ-TREE (v1.6.10) and the inbuilt ModelFinder over 1000 pseudo-bootstrap replicates, and annotated using iTOL (v5.5.1)[44–46].

**Reporting summary**. Further information on research design is available in the Nature Research Reporting Summary linked to this article.

## Data availability
The novel genome sequences generated during and analysed during the current study are available in the NCBI Genbank repository, under the study ID PRJEB20897. Source data are provided with this paper.

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

## Acknowledgements

The authors thank Prof. P.A. Hoskisson, and Z.A. Dyson for their expert knowledge and advice. This work primarily received funding from the Medical Research Council under the University of Cambridge Medical Research Council Doctoral Training Programme. Additionally, the Horizon 2020-MSCA-IF-2018; 843405-DIFTERIA. This research was supported by the NIHR Cambridge Biomedical Research Centre and AMR Research Capital Funding Scheme [NIHR200640]. The views expressed are those of the author(s) and not necessarily those of the NIHR or the Department of Health and Social Care.

## Author contributions

Conceptualisation: R.C.W., T.R.M., N.C.S., G.D. and A.M. Data curation: R.C.W. Formal analysis: R.C.W., A.K.P., K.V., and D.K. Funding acquisition: R.C.W., T.R.M., N.C.S., B.V., V.M., G.D., and A.M. Investigation: T.R.M. and N.C.S., Methodology: R.C.W., E.H., V.S., G.D., A.M. Supervision: B.V., L.S., P.H., B.D., E.H., V.M., S.B., V.S., G.D. and A.M. Validation: R.C.W. Visualisation: R.C.W. and K.V. Writing—original draft: R.C.W. Writing—review and editing: all authors.

## Competing interests

The authors declare no competing interests.
