## [Peer Review File · Nature Communications]

REVIEWERS' COMMENTS

Reviewer #1 (Remarks to the Author):

I refereed an earlier version of Will et al., and the authors have substantially addressed all of the comments I raised, adding additional data and analysis and removing some of the unsupported conclusions. The manuscript reads very well and makes a significant contribution to a relatively poorly studied organism and in an area that requires an increased research focus.

Reviewer #3 (Remarks to the Author):

The authors have addressed some of the issues raised in the previous version submitted to Nature Microbiology including the statistical support for observations made and modelling of the impact of tox mutations identified.

What is the relative contribution of recombination to mutation to the diversity observed across the species? - this question has not been addressed - it should be calculated as a ratio.

The AMR and tox variant functional analysis are still somewhat preliminary in nature (as highlighted by the authors) and overall, in depth genetic analysis is lacking.

This reviewer remains unclear about what the major new insights of the study are beyond current understanding in the literature. Importantly, the relevance of the NS tox mutations for vaccine efficacy are unclear. In fact, in the Discussion, the authors seem unclear as to whether their data 'increases confidence' in the current vaccine or suggest the possibility of future vaccine escape.

Response to Reviewers:

Reviewer #1 (Remarks to the Author):

I refereed an earlier version of Will et al., and the authors have substantially addressed all of the comments I raised, adding additional data and analysis and removing some of the unsupported conclusions. The manuscript reads very well and makes a significant contribution to a relatively poorly studied organism and in an area that requires an increased research focus.

- We thank the reviewer for all their comments and thoughts that have made the paper stronger, and for their agreement that this ID area needs urgent attention.

Reviewer #3 (Remarks to the Author):

The authors have addressed some of the issues raised in the previous version submitted to Nature Microbiology including the statistical support for observations made and modelling of the impact of tox mutations identified.

What is the relative contribution of recombination to mutation to the diversity observed across the species? - this question has not been addressed - it should be calculated as a ratio.

- It is not feasible, by any means, to correctly calculate the recombination to mutation ratio on the complete diphtheria dataset. The total recombination is the output of Gubbins screen on full genome whereas mutation based phylogeny was built using Roary derived robust core. Thus any such analysis on this would be misleading on the whole dataset. Nevertheless, to best answer this query, we have now provided recombination to mutation ratios for the two subsets (Indian and European) that were additionally analysed for studying the relative recombinational impact on clonal diphtheria lineages. This data is now added in the manuscript in lines 210 – 211 and 216 – 217.

The AMR and tox variant functional analysis are still somewhat preliminary in nature (as highlighted by the authors) and overall, in depth genetic analysis is lacking.

- The primary focus of our paper is global *C. diphtheriae* phylogeny and toxin analysis. Studying AMR mechanistically will require a separate study, which we are currently planning. Any more detail on AMR mechanisms in this paper will deviate the reader from the main vision and mission of the current study.

This reviewer remains unclear about what the major new insights of the study are beyond current understanding in the literature. Importantly, the relevance of the NS tox mutations for vaccine efficacy are unclear. In fact, in the Discussion, the authors seem unclear as to whether their data 'increases confidence' in the current vaccine or suggest the possibility of future vaccine escape.

- We have clarified this further in Lines 257 – 260 and 264 - 268